# Examination of Concrete Canvas under Quasi-Realistic Loading by Computed Tomography

**Balázs Eller** [1,2]**, Majid Movahedi Rad** [1]**, Imre Fekete** [1]**, Szabolcs Szalai** [1,*]**, Dániel Harrach** [1]**, Gusztáv Baranyai** [1]**, Dmytro Kurhan** [3]**, Mykola Sysyn** [4] **and Szabolcs Fischer** [1,*]

1    Central Campus Győr, Széchenyi István University, H-9026 Győr, Hungary
2    Faculty of Engineering and Information Technology, University of Pécs, Boszorkány u. 2, H-7624 Pécs, Hungary
3    Department of Transport Infrastructure, Ukrainian State University of Science and Technologies, UA-49005 Dnipro, Ukraine
4    Department of Planning and Design of Railway Infrastructure, Technical University Dresden, D-01069 Dresden, Germany
*    Correspondence: szalaisz@sze.hu (S.S.); fischersz@sze.hu (S.F.); Tel.: +36-(96)-613-544 (S.F.)

**Abstract:** The current paper concerns the investigation of CC (Concrete Canvas), a unique building material from the GCCM (geosynthetic cementitious composite mat) product group. The material is suitable for trench lining, trench paving, or even military construction activities, while the authors' purpose is to investigate the application of the material to road and railway substructure improvement. This research was carried out to verify the material's suitability for transport infrastructure and its beneficial effects. The authors' previous study reported that the primary measurements were puncture, compression, and the parameters evaluated in four-point bending (laboratory) tests. However, based on the results, finite element modeling was not feasible because the testing of the composite material in a single layer did not provide an accurate indication. For this reason, the material characteristics required for modeling were investigated. A unique, novel testing procedure and assembly were performed, wherein the material was loaded under quasi-realistic conditions with a crushed stone ballast sample and other continuous particle size distribution samples in a closed polyethylene tube. In addition, the deformation of the material following deformed bonding was measured by computed tomography scanning, and the results were evaluated.

**Keywords:** GCCM; Concrete Canvas (CC); railway ballast; drinking water HDPE tube; computed tomography (CT)

## 1. Introduction

Electricity consumption [1–3] will soon be a significant concern for railroads because of the worldwide energy crisis [4] and the cost of energy sources [5–8]. Therefore, it is logical for railway companies to save money by reducing the overconsumption of electricity. On the other hand, this is only one side of maintaining economical railway operation (i.e., the vehicles). The other side, the railway infrastructure (permanent way, overhead wires, signaling, etc.), is also a key factor. Of course, there is always a strong connection between all the components [9–15]. This means that the worse the railway track's geometry, the higher the required traction energy demand of the vehicles, and so on. It is important to note that similar problems arise in road transport, especially with respect to electric vehicles [16]. One sub-area of this is the issue of autonomous vehicles [17]. Transport systems should be treated and analyzed as a complex and co-existing whole. However, the current paper only focuses on the longevity of railways with respect to the above-mentioned aspects.

Railway maintenance faces different challenges, especially in countries such as Hungary, where the line network is dense, the technical conditions are mostly poor, and the sources are finite. However, thanks to the European Union's monetary support [18], this

situation is improving year after year. This paper's authors are searching for different technologies for the renewal and maintenance of railways that are less frequented but constantly used. Along these lines, the sources are few, so cost-effective solutions are always needed. The main problems were summarized in [19,20], and the deterioration of railway tracks [21,22] and ballast beds have been detailed previously [23–26].

Computed tomography (CT) was employed in differing applications with respect to the behavior of cementitious structures. This is understandable since the nondestructive nature of this method allows the samples to remain undisturbed while valuable volumetric information is reconstructed. In situ measurements are just one of the methods that are characteristic of CT. In this process, a load frame is constructed for the CT machine, thus allowing the scanned part to be under load or tension while scanning. These measurements are very convenient and straightforward once the system is established and can produce invaluable results with respect to studying the behavior of concrete structures [27,28]. Furthermore, recent advances in machine learning now permit the rapid developments made in uniquely trained systems to be applied to specific problems. Understandably, this field is very diffuse; therefore, advances in the process of constructing concrete structures are emerging rapidly [29].

The authors have examined the relevant technology using geosynthetic cementitious composite mats (GCCMs) under railway loads. The research was similar to [30,31], where UBMs (under ballast mats) were examined directly under the railway ballast layer. Both studies showed no significant damage in the UBM samples under the different loadings. Thanks to the load distribution, the materials are sufficient to withstand railway loadings and stresses.

There are different types of GCCMs, but the one investigated herein is Concrete Canvas (CC) technology. It is used to solve several engineering problems such as drainage, dewatering, slope protection, or the construction of retaining walls [32]. It is an alternative to conventional concrete, reinforced by an inner 3D fiber matrix. There is a fibrous top surface on the upper side, while the other side has a PVC (polyvinyl-chloride) waterproof layer. Thanks to these qualities, it is perfect for drainage works, but some other case studies have experimented with other options [17]. CC technology's usage in railway structures is a relatively new phenomenon; no other research paper has been published on this topic, and only the current authors have begun researching this solution with respect to railway structures. In [33,34], the authors examined the material's properties in terms of different loadings, which is very important for railways. The CC material after bonding is rigid because of the cement filler. This rigidity could be a problem under special effects such as dynamic loadings, but for different reasons [17], the authors assume that it is not as large a problem as might be assumed. The material must also be examined under dynamic loadings to determine its behavior. To counterbalance its rigidity and decrease vibration and slightly decrease noise [35–37], a USP (under sleeper pad) [38] or rail dampers [39,40] could also be utilized.

According to the authors' concept, CC can achieve the same interlocking effect as geogrids, which was proven in [41–43]. In addition, it provides perfect layer separation and adequate drainage too. Previously, the authors examined a multi-level shear box, which was discussed in [44]. The ballast particles penetrate the CC layer under pressure, and bonding occurs in this state (See Figure 1). Accordingly, the thin (i.e., 13 mm thick) CC layer works in conjunction with the lower ballast. Thus, the inner shear resistance increased, while the CC did not break. The stone particles could not be removed by hand, only by a hammer. The CC layer was thinner at these locations, while the concrete "powder" was pushed to the CC's surface. In this article, the authors investigated these phenomena in a smaller sample. The results are expected to constitute comprehensive knowledge of the thinning of the CC. In addition, the abrasion of the ballast is also examined.

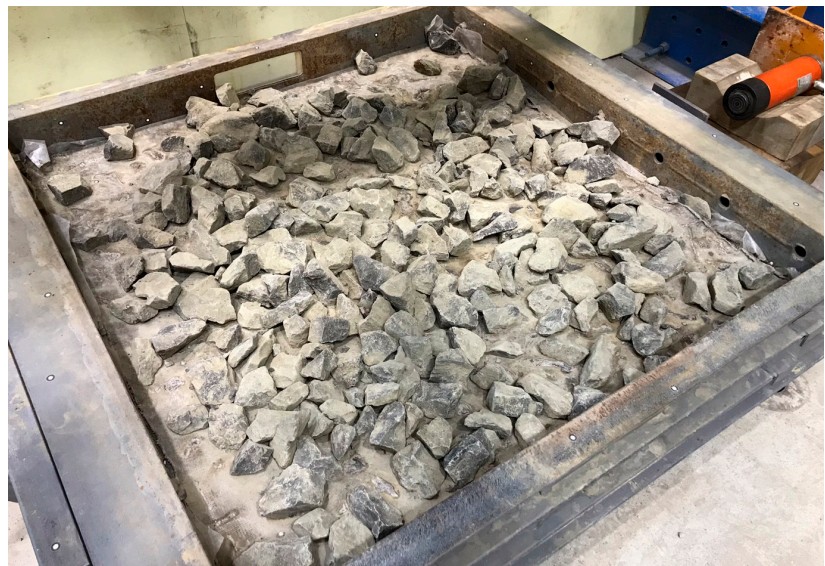

**Figure 1.** The ballast particles are cemented into the CC layer in a multi-level shear box.

## 2. Laboratory Measurements

### 2.1. Set-Up of the Measurements

The tests were performed with and without the CC layer. An HDPE (high-density polyethylene) drinking water tube with an internal diameter of 180 mm was applied for the experiment. The tubes were cut into 40 cm pieces. One side was closed by two layers of wooden plates. A total of 3-3 pieces of screws had to be screwed into the edge of the pipe in order for the XR (X-ray) data to be correctly evaluated.

To prevent the sliding of the stone particles, a 0.5 cm thick geotextile was installed on the pipe's inner surface.

In the specimens, there were three different layers. The first layer was a 10 cm thick sandy gravel layer. The moisture content was set to 5%. It was compacted in three layers with a PSC (Proctor soil compactor). The soil was compacted 25 cycles per layer. The particle size distribution of the sandy gravel can be seen in Figure 2.

**Particle size distribution - With CC layer**

Sandy gravel layer

**Figure 2.** Establishment of the inner surface and the prepared specimen without ballast particles (PSD—particle size distribution).

For the second layer, two options were planned. First, one geotextile layer was laid on the sandy gravel layer in two cases (see Figure 3). Then, one layer of CC13 (Concrete Canvas with a nominal thickness of 13 mm) was laid in three cases. Under the CC layer,

a plastic foil was installed; the execution of this step was important to prevent the water from flowing through. Otherwise, there would have been the possibility that the CC would not bind properly.

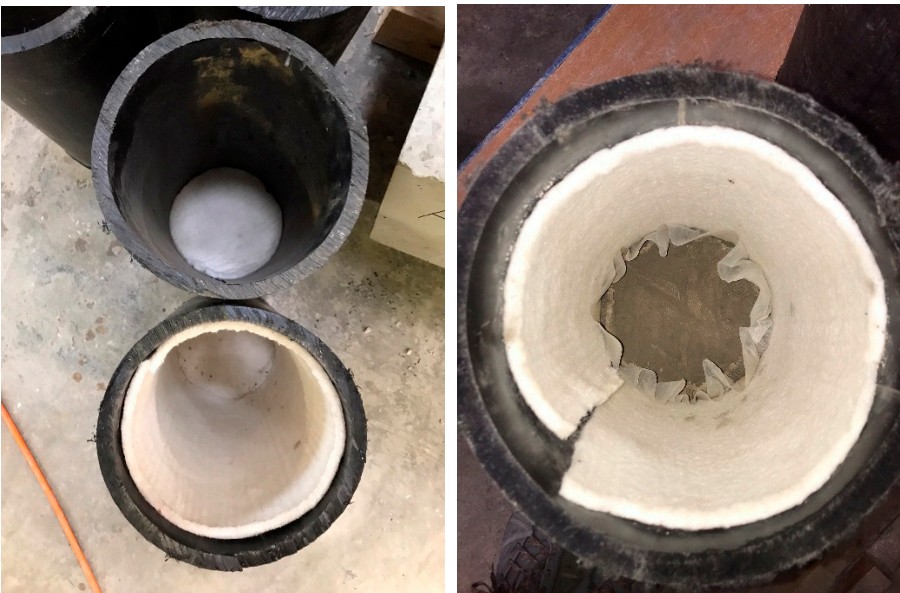

**Figure 3.** Establishment of the inner surface and the prepared specimen without ballast particles.

The Concrete Canvas had to be hydrated, but first, an initial computed tomography (CT) recording was required, which showed the initial state. Thus, there was no chance to hydrate the CC separately, so the CC was hydrated through the stone particles. According to the instructions from [32], the necessary amount of water is 9.5 L per square meter. Using the latter value as a reference, we calculated the required amount of water proportionately.

The third layer was the ballast crushed stone. The ballast material was in accordance with the MSZ EN standard [45], which is suitable for MSZ EN 13450. The standard grain size was 31.5/50 mm. $LA_{RB}$ was max. 13%, and the Micro-Deval Abrasion ($M_{DE}RB$) was max. 10%.

The particle sizes were compiled proportionately according to the tube volume for the correct particle size distribution. Approximately 5.0 . . . 5.4 kg ballast crushed stones could be installed in these tubes.

After the tube was filled with the crushed stones, it was placed on a shaking bench (vibration table). Each specimen was shaken for 60 s. In summary, the specimen's cross-section can be seen in Figure 4.

The measurement process is structured as follows:

1. After the specimen's assembly, conduct the first recording of the initial state of the specimen (via CT equipment);
2. Hydrate the specimens with CC;
3. Let the specimens with CC stand for 60 min;
4. Load with ZD-40 machine until 100 kPa is reached—hold for 5 min;
5. Conduct the second recording using CT equipment;
6. Let the specimens with CC stand for seven days;
7. Load with ZD-40 machine until 200 kPa is reached—hold for 1 min;
8. Conduct the third recording using CT equipment;
9. Measure the weight and construct the new particle size distribution graph.

The measurements needed very accurate logistics. For example, the specimens had to be loaded after 60 min, while the CT recordings also needed approximately 50–60 min. So, the assembly of the samples had to be performed in a staggered manner.

The 60 min waiting period is not an instruction or an exact requirement. It was the authors' decision according to previous experience. On the other hand, before the 3rd CT measurement, the specimens were left to stand for seven days. According to the authors' previous measurements [33], a seven-day period is sufficient to obtain 95–96% of the specimens desired properties.

The aim of the CT recordings is to measure how the ballast particles can penetrate into the CC layer and, moreover, how this affects the level of abrasion.

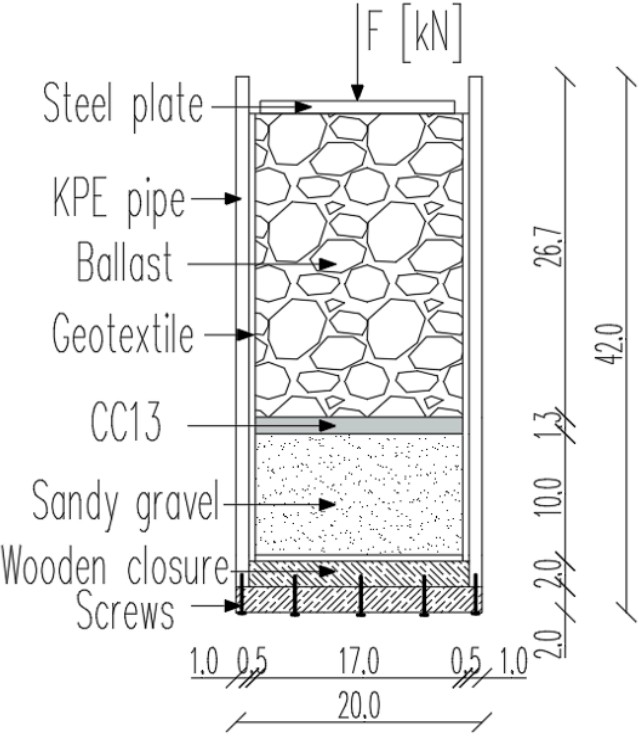

**Figure 4.** Set-up of the test specimens (KPE equals HDPE) (all the dimensions are in cm unit).

*2.2. Computed Tomography Examination*

The idea presented herein came from [46], but there are many publications about 3-dimensional image analysis, such as [47–49].

The device used in this study was provided to the authors by the laboratory of Audi Hungaria Faculty of Automotive Engineering at Széchenyi István University, Győr (Hungary).

According to [46], the basic specifications of the device and some relevant data to perform one of the measurements are as follows:

- A 360° rotation produces 1260 projections (CT images);
- The number of lines is 104;
- In the case of multi-slice, the distance between two slices is 210 mm;
- The number of pixels is 2048 × 2048 (used: 1024 × 1024);
- 2D-pixel size: 0.19124188 mm;
- 3D-XY-pixel size: 0.18966927 mm (the edge length of 1 spatial pixel, the so-called 'voxel');
- 3D-Z-pixel size: 0.1896692 mm;
- X-ray tube: Y.TU 450-D09;
- Tube voltage: 0 . . . 450 kV (used 210 kV);
- Current: 2.60 mA (this is paired with 210 kV, e.g., 1.213 mA for 450 kV);
- Focus: small;
- Filter:

　○　　Al—0.00 mm;
　○　　Cu—1.50 mm;
　○　　Sn—0.00 mm;
　○　　Pb—0.00 mm.

Deformation and thinning of the CC and movement of the railway ballast were investigated utilizing computed tomography. The scans were performed in a YXLON Modular Y.CT scanner using a mini-focus X-ray tube and a flat-panel detector. The parts were scanned at 450 kV tube voltage and 1.5 mA tube current. The detector was operated at 1000 ms integration time to capture the images. A total of 1440 projections were made and reconstructed into a voxel grid with a voxel size of 0.207 mm. Therefore, the accuracy of an individual scan is 0.207 mm. Figure 5 shows the CT setup with the detector on the left capturing the images and the X-ray tube on the right generating the conical beam of X-rays. Between the test cylinder sitting on the rotary table that revolves around during measurement. Note that a second X-ray tube is present (top right) in this setup but was not used for this study due to its low power.

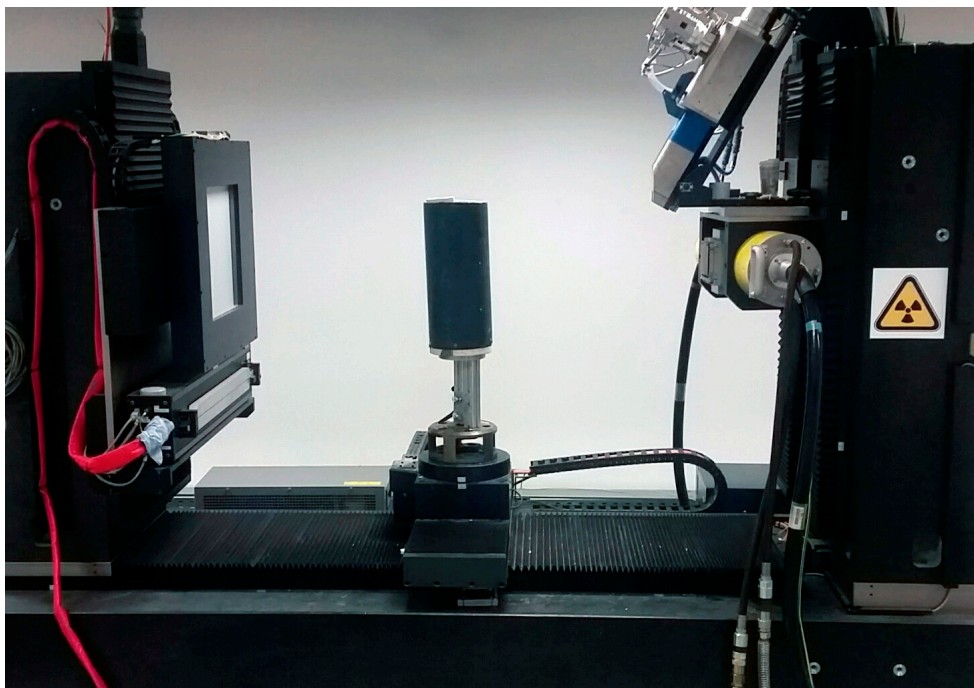

**Figure 5.** From left to right: Computed tomography setup with a flat panel detector, the test cylinder, and the X-ray tube.

In order to allow for the registry of the reconstructed data in the same coordinate system, steel ball bearings were embedded in the walls of the polymer test tube. The ball bearings did not affect the testing process or the quality of the CT scan. The test cylinders were scanned in three stages, as discussed in Section 2.1. First, the reconstructed data were processed using VGStudio MAX. The initial CT scan was the reference for the best-fit registration of the remaining scans. The best-fit registration used the ball-bearing positions to align the scans to the reference scan. Figure 6 shows slices from the three different stages with and without CC. Close inspection of these slices reveals the slight movement of the materials. However, these differences are minor; therefore, reliance on visual inspection would be inaccurate. Different measurements of the scans were taken in order to gather data on the behavior of the CC and the ballast particles. However, a problem occurred after shaking the samples. The placement of the geotextile was not appropriate; the sandy gravel was pushed up on the sides.

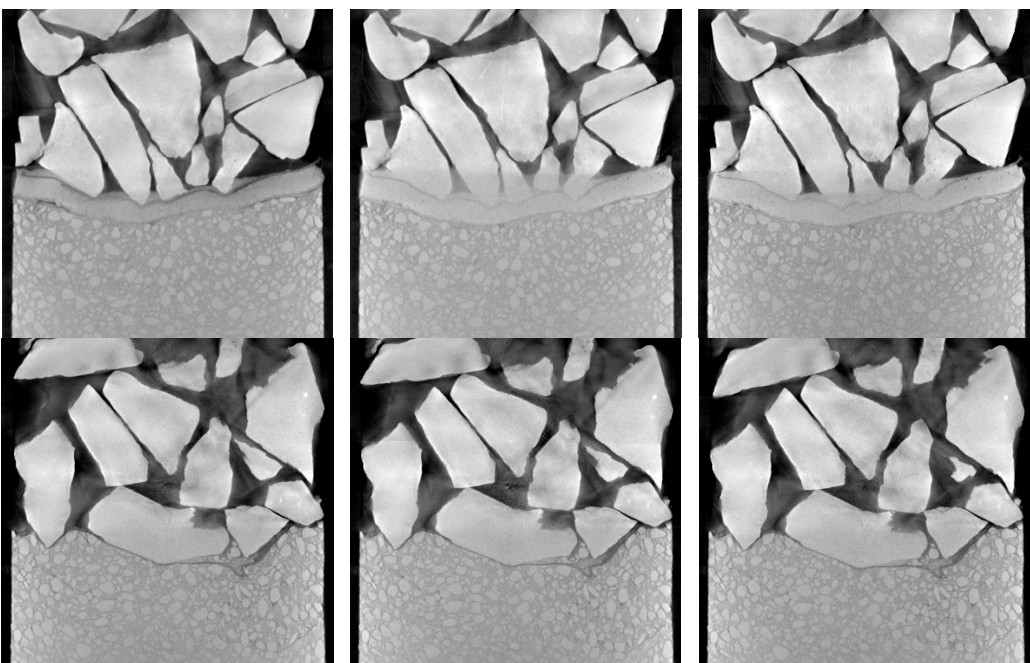

**Figure 6.** Typical test tube reconstructions with (**top** row) and without (**bottom** row) a CC layer. Different stages are shown from (**left**) (see the Section 2.1 above Figure 4, point #2), through (**middle**) (see the Section 2.1 above Figure 4, point #5) to (**right**) (see the Section 2.1 above Figure 4, point #8).

One such measurement was to determine the thickness change of the CC layer. A region of interest was selected to cover the CC within the scan. This process was performed using selection tools and modules in the software employed in this study. The thickness was determined by sending measurement lines oriented orthogonally to the current surface to the opposite side of the region of interest (ROI).

Digital volume correlation (DVC) was employed in order to visualize the displacements of the features of the scans, such as the movement of the ballast, CC, and sandy gravel. DVC functions by matching easily identifiable regions on two sets of reconstructed volumes. These regions are usually edges, corners, and small, non-homogeneous features of the object. Correctly matching these features in the original and the deformed volume can determine the strain and deformation. Vectors or colormaps can visualize these displacements and strain fields.

The ballast's displacement was investigated in three different regions by determining an individual ballast's center mass. The position of the CC after each loading stage reveals the movement of the ballast.

## 3. Results

### 3.1. Abrasion and Breakage of the Ballast Material

One of the determined parameters is the particle size distribution measured before and after the test. The authors assumed that the CC provides enough rigidity such that the elasticity decreases on the crown, which increases the degradation of the ballast crushed stones. The particle size distributions can be seen in Figure 7 (without a CC layer) and Figure 8 (with a CC layer).

It can be seen on the graphs that the degradations were not so great under these two loadings, but occurred unequivocally, nonetheless.

It was found that the degree of the degradation of the ballast crushed stones is nearly the same in the two cases. Unfortunately, the standard deviation of the abrasion was considerable, so it is impossible to find a clear difference in these small samples.

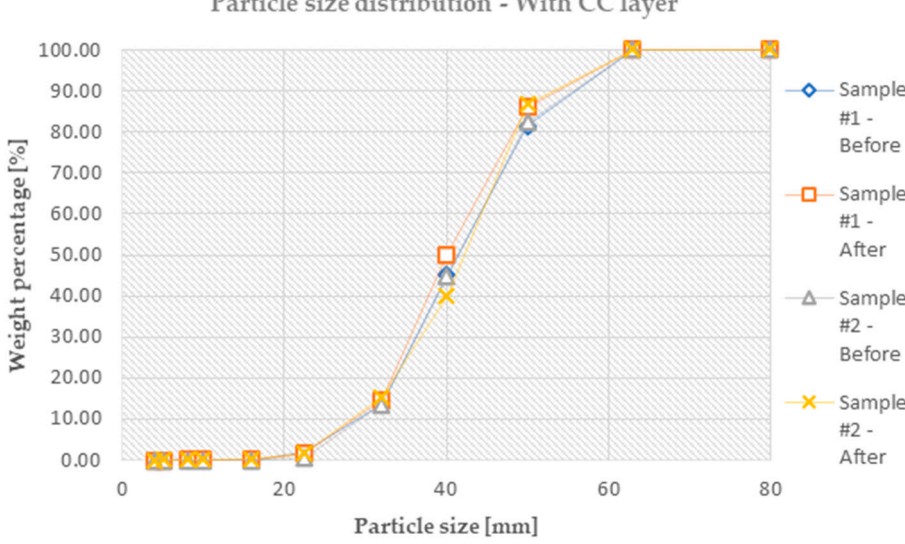

**Figure 7.** The particle size distribution before and after the tests for the specimens without CC layer (PSD—particle size distribution).

Particle size distribution - With CC layer

**Figure 8.** The particle size distribution before and after the tests for the specimens with CC layer (PSD—particle size distribution).

In Figure 8, only two samples can be seen. Earlier, it was mentioned that the particles could bind to the CC layer. This could also be a problem during the evaluation because the particles cannot be removed by hand from the CC layer. In these cases, a hammer has to be used, which causes the particles to break into the layer. This happened in Sample #3. The particle size distribution was not correct; two of the particles were broken into smaller pieces thanks to the aggressive removal process.

The authors supposed that the loading speed can also be a relevant parameter, which will be measured in the following investigations.

### 3.2. Thinning of the CC under Pressure

The measurement of these data comes from geogrids, which have been proven to constitute a good solution for railway ballast stabilization. In Section 3.1, it was mentioned that the particles could bind to the CC layer. This is crucial because the CC layer and the bonded crushed stone particles collaborate in these cases. The bonded ballast particles are

similar to those interlocked in the geogrids. Therefore, the friction of the layer increases significantly, which also helps form an interlocking effect. The bonded particles can be seen in Figure 9.

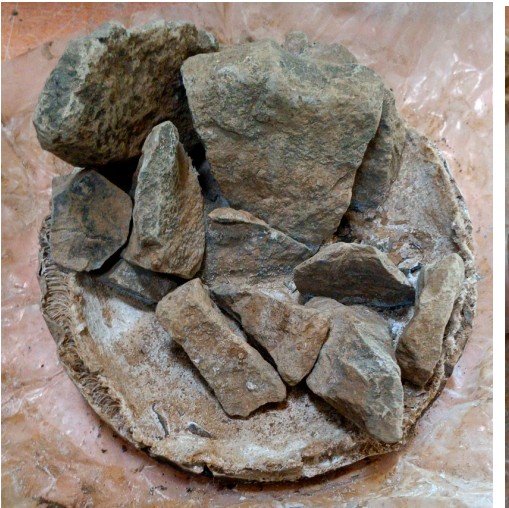
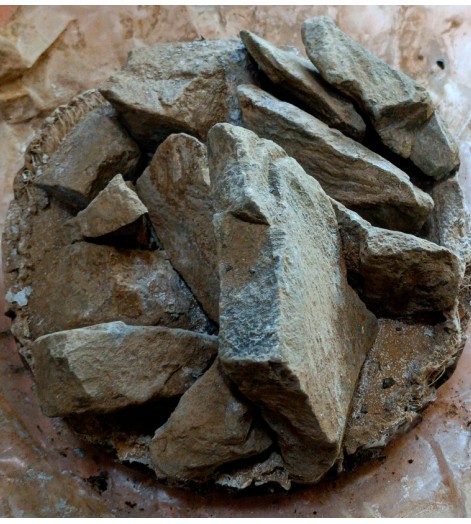

**Figure 9.** The ballast particles bonded in the CC layer. These stones could not be removed by hand.

The thickness measurements of the CC based on the CT reconstruction revealed differing amounts of thinning. Figure 10 shows a typical thickness profile of the CC. The thickness profiles were obtained by determining the CC's top and bottom surfaces and executing the thickness analysis provided by the analysis software. The thickness analysis is performed by projecting lines from one surface to the opposing surface and measuring the length of the projected line segment. These lengths correspond to the thickness of the two surfaces. The thickness is presented using a colormap overlaid on the CC. Warmer colors (red and yellow) correspond to thick regions close to the original thickness of the CC. In contrast, colder colors (cyan and blue) represent thin regions.

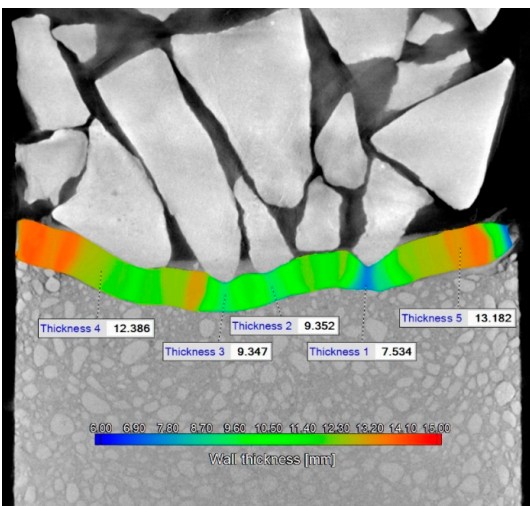

**Figure 10.** Thickness measurement of CC.

A great degree of thinning occurred where the ballast is oriented such that a pointed feature faces the CC (Thicknesses 1 to 3 in Figure 10). These regions have a thickness value in the range of 10 . . . 7 mm, which is highly dependent on the shape of the ballast as well as its position. A small amount of thinning is present where the ballast rests on the CC with a flat face. In this case, the thickness is close to the original thickness of the CC, in

the range of 12 . . . 10 mm (Thickness 4 in Figure 10). Where no ballast is contacting the surface of the CC, the original thickness could be measured (Thickness 5 in Figure 10). In this case, the measured thickness was 13.1 mm, close to the nominal 13.0 mm thickness. This measurement on the non-deformed CC compared to the nominal thickness reveals the approximate error of the measurement, which is around ±0.1 mm.

Due to the lower support and the CC layer's composition, the layer was not perforated under the sharp stone particles.

The digital volume correlation results are presented in Figure 11, which displays the displacements along the axis of the test cylinder. DVC produces strain and displacement in three directions; however, in this case, the relevant direction is parallel to the Z-axis, which is the loading direction. Typical results are presented for the test cylinders with CC and without CC. The presented results show the Z direction displacement field color mapped to the surface of the contents of the test cylinder. The color map displays the movement from the original to the loaded stage. Warmer colors represent a slight movement downward or upward, while colder colors denote a large degree of downward movement. The most apparent displacements are located at the top of the ballast layer. Here, the rocks are the most loosely packed; therefore, movement is not restricted. However, due to the compression, this movement is likely to occur in the negative direction, as seen by the blue colors. In some cases, the movement is constrained, and a slight rotation occurs, which causes parts of the ballast to move up.

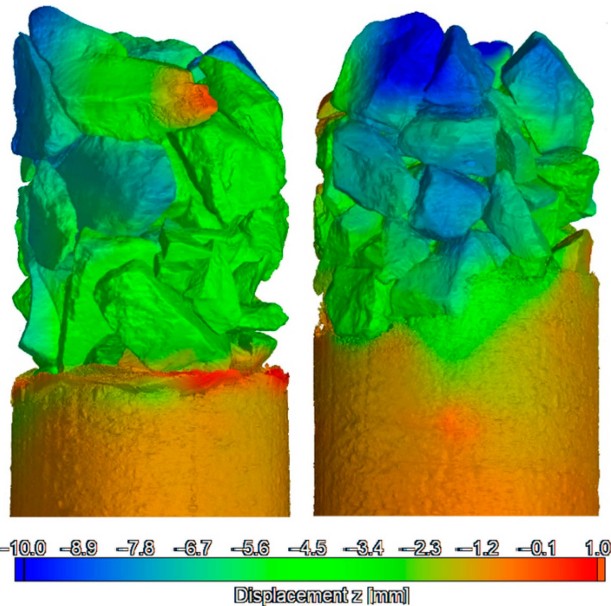

**Figure 11.** Displacements of the materials in the test cylinders with CC and without a CC layer ((**left**) and (**right**), respectively).

Lower in the ballast, the displacements are less significant since this area is more closely packed, thus not allowing free movement.

The sandy gravel has a uniform level of displacement in the test tube that contains CC. Whereas in the test tube without CC, only containing geotextile, the displacement of the sandy gravel is only uniform at the lower regions. Closer to the ballast, the sandy gravel is pushed slightly upward by the ballast, as seen in the yellow-colored region.

It can be seen clearly in Figure 11 that on the right-side sample, the geotextile moved onto the edge, which disrupted the layer separation. On the other hand, the CC layer ensured adequate separation.

The displacement of an individual ballast particle was measured by determining the ballast's center mass and the center of mass's movement over the different stages. Figure 12

shows the outline of the ballast overlaid from the three different loading stages, as well as the trajectory of the center mass (shown in yellow).

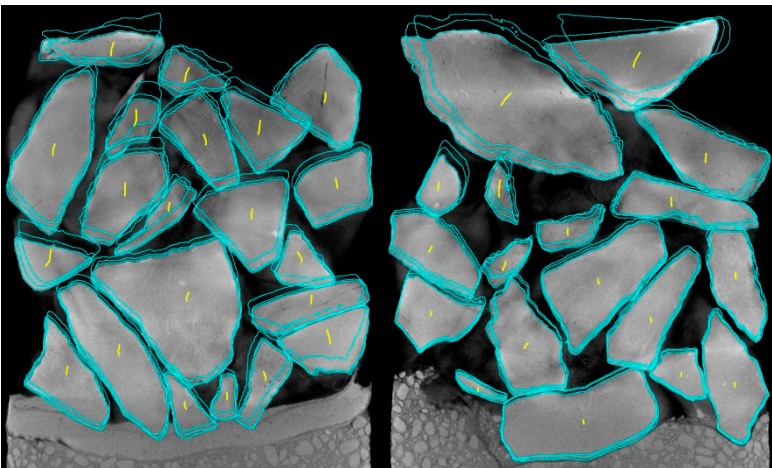

**Figure 12.** Outline and center-of-mass trajectory of the ballast with CC and without a CC layer ((**left**) and (**right**), respectively).

The top ballast layer exhibited the highest degree of movement since it constituted the most loosely packed stones. For both types of test cylinders, the movement of the top ballast layer was (on average) 11.3 mm, with the largest displacement being 27.8 mm, while the smallest was 4.8 mm. The lowest layer of ballast closest to the CC and the sandy gravel revealed differing results. On average, the lowest layer of ballast above the CC moved 4.1 mm, with the greatest movement of 5.2 mm and the least amount of movement of 3.5 mm. Whereas the ballast resting on the sandy gravel moved an average of 2.6 mm with a narrow minimum and maximum movement range of 1.8 mm and 3.2 mm, respectively. The markedly more significant movement of the ballast on the CC layer can be attributed to the flexible nature of the CC before setting. This allowed the ballast to undergo more significant movement, unlike the sandy gravel that supported the ballast slightly more. Due to this flexible behavior, the ballast particles could penetrate the CC layer, which was bound in this state.

## 4. Discussion

This research aims to examine the connection between ballast particles and Concrete Canvas. It was proven that the ballast particles penetrated the CC layer and were bound in this state. One of the questions concerned changing the thickness of the CC and how this can affect the layer's behavior. It was also inquired as to whether the CC layer was perforated or not. The results show that when the CC layer has regular lower support, perforation will not occur. On the other hand, the penetrated ballast stone particles push out the cement from the layer structure, and the cement bonds on the surface of the CC layer. In these cases, these stone particles can not be removed by hand.

The second necessary behavior that the authors studied was ballast breakage. In the research, the samples were loaded by quasi-realistic static loading (100 and 200 kPa). It can be stated that the rigidity of the CC can result in more considerable rigidity in the lower ballast too, in which more extensive abrasion occurs. According to the results, the levels of degradation of the ballast particles were not very different. The samples with CC and without CC layers had nearly the same levels of abrasion. Although calculating the standard deviation was ineffective, the results were significant; larger samples are needed for a more accurate result.

This research further strengthens the thesis that the CC layer could be an adequate protection layer on the crown of the railway subgrade, which could function as strengthening, barrier, and dewatering layers.

## 5. Conclusions

The CT investigation was remarkably successful, and the results were explanatory. The experiments performed in this study showed once again that the Concrete Canvas layer could ensure an adequate protective layer in the structure. If the particles are able to penetrate the layer, they can perform better.

The following conclusions can be stated based on the results of the present paper:

- The rigidity of the Concrete Canvas does not increase abrasion under high static loads.
- The thickness of the CC layer decreased if the ballast particles penetrated it, but the layer was not perforated. If there is support (for example, the sandy gravel layer) under the CC layer, it takes the shape of the developed crown plate.
- The laboratory tests showed the same results as in the shear box tests. One hour after hydration, the Concrete Canvas was applied with 100 kPa for 5 min. Consequently, the ballast particles bonded in the canvas. After this, this the CC layer and the bound crushed stone particles formed one layer and worked together under loading.

Future studies will investigate the nature of the CC layer under dynamic loadings. These results are auspicious because the CC receives a dead load from the railways while it takes the shape of the subgrade's crown. This means that bending cannot happen, so the CC's rigidity will not be a problem. On the other hand, if bending occurs because of sinking (of the settlement), the degree of the former cannot be so large that it breaks the material, and thus the level of drainage can still be adequate.

**Author Contributions:** Conceptualization, B.E. and S.F.; methodology, B.E. and S.F.; software, B.E., I.F. and S.F.; validation, B.E., I.F. and S.F.; formal analysis, B.E., I.F., S.S., D.H. and G.B.; investigation, B.E., M.M.R., I.F., S.S., D.H., G.B., D.K., M.S. and S.F.; resources, B.E., M.M.R., I.F., S.S., D.H., G.B. and S.F.; data curation, B.E. and I.F.; writing—original draft preparation, B.E., M.M.R., I.F., S.S., D.H., G.B., D.K., M.S. and S.F. and S.F.; B.E., M.M.R., I.F., S.S., D.H., G.B., D.K., M.S. and S.F.; visualization, B.E., I.F., and S.F.; supervision, M.M.R., S.S., D.K., M.S. and S.F.; project administration, B.E. and S.F.; funding acquisition, S.F. All authors have read and agreed to the published version of the manuscript.

**Funding:** This research received no external funding.

**Data Availability Statement:** Not applicable.

**Acknowledgments:** This paper was prepared by the research team "SZE-RAIL". The authors would like to thank E-Tronics Kft. Hungary for their help in allowing the analysis of the reconstructed CT data. The railway ballast was given by Colas Hungary Ltd., from the quarry of Szob, Hungary. The CC layers were bought from the Concrete Canvas with the help of Sodego Consulting.

**Conflicts of Interest:** The authors declare no conflict of interest.

## Abbreviations

| | |
|---|---|
| 2D | two dimensions or two-dimensional |
| 3D | three dimensions or three-dimensional |
| CC | Concrete Canvas |
| CT | computed tomography |
| DVC | digital volume correlation |
| GCCM | geosynthetic cementitious composite mat |
| HDPE | high-density polyethylene |
| PSC | Proctor soil compactor |
| PSD | particle size distribution |
| PVC | polyvinyl-chloride |
| ROI | region of interest |
| UBM | under ballast mat |
| USP | under sleeper pad |
| XR | X-ray |

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
