# Peer review of "Examination of Concrete Canvas under Quasi-Realistic Loading by Computed Tomography"

_infrastructures, doi:10.3390/infrastructures8020023_

Round 1

Reviewer 1 Report

The authors have carried out a unique new test procedure and assembly so that under the quasi-real conditions, the material is loaded into the gravel ballast sample and other continuous particle size distribution samples in the closed polyethylene pipe, and the deformation of the material after deformation and bonding is measured by computer tomography, and the results are evaluated. The research content of this paper is relatively new. Some suggestions are as follows:

1. It is suggested to list the physical schematic diagram of CT scanning so that readers can understand the scanning process more clearly.

2. It is recommended to list parameters such as CT scanning accuracy.

3. It is suggested to explain the specific meaning of different colors in Figure 9, that is, the specific meaning of the color corresponding to Wall thickness in the legend. And explain how to get these.

4. It is suggested to explain the specific meaning of different colors in Figure 10. That is, the specific meaning of the color corresponds to Displacement in the legend. And explain how to get these.

5. It is recommended to refer to relevant documents, such as:

1Identification and reconstruction of concrete mesostructure based on deep learning in artificial intelligenceConstruction and Building MaterialVolume 352, 17 October 2022, Jingwei Ying Jiashuo Tian Jianzhuang Xiao Zhiyun Tan 

2X-ray computed tomography images based phase-field modeling of mesoscopic failure in concreteEngineering Fracture Mechanics

3In-situ X-ray computed tomography characterization of 3D fracture evolution and image-based numerical homogenization of concreteCement and Concrete CompositesVolume 75, January 2017, Pages 74-83

Author Response

See the attached PDF file.

Reviewer 2 Report

I suggest improving the graphical quality of the drawings and their descriptions (e.g. Fig. 1 and 8).

The article presents the parameters of the computed tomography measuring system in detail . Similar parameters should be provided for the structure of the analyzed track.

1) What are the parameters of the ballast used in the tests presented in the article (according to EN 13450)?

a) what is the grain size: 31,5/50 or 31,5/63 ?

b) what MDE RB ?

c) what LARB  ?

2) I suggest moving the text: “The sample was given by Colas Hungary Ltd., from the quarry of Szob, Hungary.” from the article to the Acknowledgments.

3) The authors write: "To be able to counterbalance its rigidity and decrease vibration and slightly the noise [28–30], e.g., USP (under sleeper pads) [31] or rail dampers [32,33] could also be utilized." Meanwhile, the location of the rail damper in the track construction is completely different than Concrete Canvas.

Please supplement the “Introduction” with research on UBM mats, which, like Concrete Canvas, are located directly under the ballast layer. UBM mats are also tested for damage from sharp bedding grains (in a ballast box or using a GBP plate). UBM may have an impact on the stress in ballast, as presented in the works - for example:

1) A. de O. Lima, M. S. Dersch, Y. Qian, E. Tutumluer & J.R. Edwards Laboratory mechanical fatigue performance of under-ballast mats subjected to North American loading conditions

2) https://www.mdpi.com/1996-1944/14/9/2125

Author Response

See the attached PDF file.
